# Incorporation of Oxidized Phenylalanine Derivatives into Insulin Signaling Relevant Proteins May Link Oxidative Stress to Signaling Conditions Underlying Chronic Insulin Resistance

**DOI:** 10.3390/biomedicines10050975

**Published:** 2022-04-22

**Authors:** Judit Mohás-Cseh, Gergő Attila Molnár, Marianna Pap, Boglárka Laczy, Tibor Vas, Melinda Kertész, Krisztina Németh, Csaba Hetényi, Orsolya Csikós, Gábor K. Tóth, Attila Reményi, István Wittmann

**Affiliations:** 12nd Department of Medicine and Nephrology-Diabetes Center, University of Pécs Medical School, 7624 Pécs, Hungary; mohas-cseh.judit@pte.hu (J.M.-C.); molnar.gergo@pte.hu (G.A.M.); laczy.boglarka@pte.hu (B.L.); vas.tibor2@pte.hu (T.V.); meli871106@gmail.com (M.K.); 2Department of Medical Biology and Central Electron Microscopic Laboratory, University of Pécs Medical School, 7643 Pécs, Hungary; pap.marianna@pte.hu; 3Signal Transduction Research Group, Szentágothai Research Centre, University of Pécs, 7624 Pécs, Hungary; 4Institute of Organic Chemistry, Research Centre for Natural Sciences, 1117 Budapest, Hungary; nemeth.krisztina@ttk.hu (K.N.); remenyi.attila@ttk.mta.hu (A.R.); 5Department of Pharmacology and Pharmacotherapy, University of Pécs Medical School, 7643 Pécs, Hungary; hetenyi.csaba@pte.hu; 6Department of Medical Chemistry, Albert Szent-Györgyi Medical School, University of Szeged, 6725 Szeged, Hungary; csikos.orsolya@icloud.com (O.C.); toth.gabor@med.u-szeged.hu (G.K.T.)

**Keywords:** insulin resistance, oxidative stress, hydroxyl free radical, ortho-tyrosine, meta-tyrosine, IRS-1, phosphorylation, dephosphorylation

## Abstract

A link between oxidative stress and insulin resistance has been suggested. Hydroxyl free radicals are known to be able to convert phenylalanine (Phe) into the non-physiological tyrosine isoforms ortho- and meta-tyrosine (o-Tyr, m-Tyr). The aim of our study was to examine the role of o-Tyr and m-Tyr in the development of insulin resistance. We found that insulin-induced uptake of glucose was blunted in cultures of 3T3-L1 grown on media containing o- or m-Tyr. We show that these modified amino acids are incorporated into cellular proteins. We focused on insulin receptor substrate 1 (IRS-1), which plays a role in insulin signaling. The activating phosphorylation of IRS-1 was increased by insulin, the effect of which was abolished in cells grown in m-Tyr or o-Tyr media. We found that phosphorylation of m- or o-Tyr containing IRS-1 segments by insulin receptor (IR) kinase was greatly reduced, PTP-1B phosphatase was incapable of dephosphorylating phosphorylated m- or o-Tyr IRS-1 peptides, and the SH2 domains of phosphoinositide 3-kinase (PI3K) bound the o-Tyr IRS-1 peptides with greatly reduced affinity. According to our data, m- or o-Tyr incorporation into IRS-1 modifies its protein–protein interactions with regulating enzymes and effectors, thus IRS-1 eventually loses its capacity to play its role in insulin signaling, leading to insulin resistance.

## 1. Introduction

### 1.1. Pathogenesis of Insulin Resistance

The pathogenesis of type 2 diabetes is complex [1,2], but one of the hallmarks of the development of type 2 diabetes—at least in obese patients with type 2 diabetes—is peripheral chronic insulin resistance, with a reduced uptake of glucose into adipose tissue [3]. The triggering factor for insulin resistance is oxidative stress due to systemic subclinical inflammation and hormonal interactions which all induce oxidative stress by the activation of nicotinamide nucleotide (NAD(P)H) oxidase enzyme [4]. In obesity, overfeeding may lead to glycotoxicity and lipotoxicity. This is especially the case for the liver, as non-alcoholic fatty liver disease is a frequent companion to obesity. The affected tissues may face serious damage, and with time, exhaustion of the mitochondria. The strong beta-oxidation and uncoupling of the production of adenosine triphosphate (ATP) and terminal oxidation may lead to a higher rate of the formation of reactive oxygen species (ROS), contributing to the development of insulin resistance [5].

### 1.2. Acute Insulin Resistance

Acute insulin resistance induced by tumor necrosis factor alpha (TNFα) or dexamethasone could be ameliorated by the antioxidants N-acetylcysteine, manganese (III) tetrakis (4-benzoicacid) porphyrin (MnTBAP), or by induction of antioxidant enzymes such as superoxide dismutase or catalase [6]. These data suggest a strong connection between oxidative stress and the development of acute insulin resistance.

### 1.3. Chronic Insulin Resistance

In clinical settings, chronic antioxidant therapy of patients (e.g., in the case of diabetic neuropathy using alpha-lipoic acid) is able to improve chronic insulin resistance [7]. Moreover, using the so-called “breakthrough” therapy, a 2–4 week decrease in glucotoxicity is able to normalize glycemia by decreasing insulin resistance for months or years [8,9]. Because clinical insulin resistance can be reverted, it is not an alteration of the DNA, but rather some type of protein abnormality; this can be hypothesized given the background of this oxidative stress-induced clinical condition.

This theory is further augmented by the observation that a low advanced glycation end products (AGE) diet, which is associated with a state of increased oxidative stress and subclinical inflammation, ameliorated insulin resistance in obese people with the metabolic syndrome [10].

Oxidative stress may develop in type 2 diabetes mellitus because of the activation of the polyol pathway leading to a depletion in the glutathione pool, non-enzymatic glycation, and interaction of the resulting AGEs with their receptors (receptor for AGE, RAGE), which in turn may augment the same pro-oxidant NAD(P)H oxidase (NOX) via protein kinase C signaling and the hexosamine pathway. Moreover, superoxide overproduction in the mitochondria due to hyperglycemia may activate all of these pathways, resulting in a vicious cycle [11] (Figure 1).

### 1.4. Tyrosine Isomers and Hydroxyl Free Radical

Reactive oxygen species (ROS) that arise from oxidative stress processes are highly reactive and may attack macromolecules, such as lipids, nucleic acids, proteins, and amino acids [34]. Stable products of ROS-derived macromolecular damage may be used as markers of oxidative stress.

Phenylalanine (Phe) is an essential amino acid that is further used for the production of para-tyrosine, dihydroxy-phenylalanine (DOPA), catecholamines, melanine, and thyroid hormones [35]. Beyond these important enzymatic reactions, Phe, due to the vulnerability of its aromatic ring, may also be a subject of non-enzymatic oxidation processes, i.e., the attack of ROS [36]. The isomers of the physiological para-tyrosine (p-Tyr), namely meta- and ortho-tyrosine (m-Tyr and o-Tyr), are formed this way [36] (Figure 2).

### 1.5. Non-Physiological Tyrosine Isomers as Markers

Elevated levels of m- and/or o-Tyr have been described in the vascular wall of cynomolgus monkeys [37] as well as in cataract lenses [38]. In previous studies by our group, the non-physiological Tyr isomers (m- and o-Tyr) have been shown to be oxidative stress markers in patients with type 2 diabetes with or without chronic kidney disease [39] in patients with end-stage renal disease or patients with severe sepsis. We have also shown that type 2 diabetic patients treated with resveratrol showed a decrease in urinary excretion of o-Tyr, and a concomitant improvement in insulin resistance [40]. These data suggest that m- and o-Tyr may be used as markers of oxidative stress.

### 1.6. Non-Physiological Tyrosine Isomers as “Makers”

Furthermore, m- and o-Tyr may have a role beyond being markers ROS attack. Namely, m-Tyr has been shown to act as a natural herbicide and inhibit the growth of plants [41]. Additionally, m-Tyr seems to be involved in the inhibition of concomitant tumor growth [42]. These data suggest that m-Tyr is more than just a marker, but it also plays a pathogenetic role under some circumstances. For example, we have described that patients on renal replacement therapy show a connection between plasma levels of o-Tyr and erythropoietin (EPO) resistance. In a further set of experiments, m- and o-Tyr inhibited the EPO-dependent proliferation of erythroblasts, in a time- and dose-dependent manner [43]. 

With all of the above-mentioned data taken into account, we hypothesized that m- and o-Tyr—on the basis of their potential role in EPO-resistance—are able to induce chronic insulin resistance in fat cells, HEK cells, podocytes, and macrophages, and that intracellular signaling of insulin may be disrupted this way. 

Thus, proof-of-concept experiments were designed to address this question. Our results suggest that oxidative stress-induced o- and m-Tyr could incorporate into cellular proteins, interfere with insulin signaling, inhibit glucose uptake, and thus induce chronic insulin resistance.

## 2. Materials and Methods

### 2.1. Cell Culture

Early passages of mouse embryo fibroblast (3T3-L1) (ATCC, Manassas, VA, US) were cultured in Dulbecco’s modified Eagle medium (DMEM, Sigma Aldrich, Budapest, Hungary, CAT number: D6046; Invitrogen, Waltham, MA, USA, CAT number: 41966-029) supplemented with 100 U/mL penicillin, 0.1 mg/mL streptomycin (Gibco, Budapest, Hungary, CAT number: 15070-063) and 10% heat-inactivated Fetal Bovine Serum (FBS, Gibco, CAT number: 16170-078), supplemented with 398 nM p-, m-, or o-Tyr (equimolar to the original p-Tyr content of the medium) purchased from Sigma (CAT number: para-Tyr: T8566, m-Tyr: T3629, o-Tyr: 93851). Media contained 25 or 5 mmol/L glucose (experiment dependent) pyruvate and L-glutamine. Cells were grown at 37 °C and 5% CO_2_. Adipocyte differentiation was achieved by DMEM supplemented with 10% FBS (Gibco, Csertex, Budapest, Hungary, CAT number: 10106-169) (FBS) and a 0.17 nmol/L insulin (Sigma Aldrich, Budapest, Hungary, CAT number: I 9278), 0.5 nmol/L isobuthylmethylxanthine (Sigma Aldrich, Budapest, Hungary, CAT number: I 5879), and 250 nmol/L dexamethasone (Sigma Aldrich, Budapest, Hungary, CAT number: 861871) containing cocktail. From day 4 onward, cultures were maintained in DMEM containing 1.5 μg/mL insulin and 10% FBS with a medium change every other day until experimental treatments were started. After 90% of the cell population reached the adipocyte phenotype, prior to all treatments, cells were incubated overnight in serum-deprived medium. Experimental treatment was performed in serum-deprived medium containing 200 or 400 nmol/L insulin, for 5 min.

The conditionally immortalized human podocyte cell line (provided by Moin Saleem, University of Bristol, UK) was cultured in RPMI1640 medium (R8756, Sigma Aldrich, Budapest, Hungary) supplemented with 10% heat-inactivated FBS (16170-078, Gibco, Budapest, Hungary), insulin-transferrin-selenium supplement (41400-045, Gibco), 100 U/mL penicillin, 0.1 mg/mL streptomycin (P4333, Sigma), and 112.5 nM (equimolar to the original p-Tyr content of the medium) para-, meta-, or ortho-Tyr. The cells were grown at 33 °C 5% CO_2_, and when they reached 40–60% confluency they were transferred to 37 °C to differentiate. Following the thermoswitching, cells were kept on RPMI1640 medium containing 2% FBS, antibiotics, and different tyrosines. Before every treatment, cells were incubated in serum-deprived medium overnight. 

HEK-293 immortalized cell line (ATCC, Manassas, VA, USA) of epithelial morphology and the J774A.1 mouse BALB/C monocyte-macrophage cell lines (91051511-1VL, Sigma Aldrich, Budapest, Hungary) were cultured in Petri dishes with a medium composed of Dulbecco’s modified Eagle’s medium (DMEM, Sigma Aldrich, Budapest, Hungary, CAT number: D6046, Invitrogen, CAT number: 41966-029), with a 10% heat-inactivated FBS supplementation (16170-078, Gibco), insulin-transferrin-selenium supplement (41400-045, Gibco, Budapest, Hungary,), 100 U/mL penicillin and 0.1 mg/mL streptomycin (P4333, Sigma) and 398 nM para (5 and 25 mmol/L), meta-, or ortho-Tyr (T8566, T3629, 93851, Sigma). Before treatment, cells were incubated in serum-deprived medium overnight as well. After the treatment, cells were washed twice with saline (4 °C) and then exposed to 80 μL lysis buffer/plate, containing 1 mol/L Trisbase, pH 7.4, 1.15% Triton X, 0.2 mol/L EGTA, pH 7, 0.5 mol/L EDTA, pH 8, 5 mg/mL phenylmethylsulfonyl fluoride (PMSF), 0.1 mol/L dithiothreitol (DTT), 0.1 mol/L Na_3_VO_4_, 5 mg/mL aprotinin, 5 mg/mL leupeptin, and phosphatase inhibitor cocktails 1 and 2 (Sigma Aldrich, Budapest, Hungary, CAT number: P5726 and P2850). Cells were scraped off mechanically and then were frozen at −70 °C.

### 2.2. Isotope Glucose Uptake

Initially, cells were kept in glucose-free DMEM (Gibco, Csertex, Budapest, Hungary) for 30 min and then were treated with 2, 20, 200, or 400 nmol/L insulin for 100 min. Concurrently, 1 μCi/mL deoxy-D-glucose 2-[1 2-3H(N)] (3.7 × 104 Bq/mL) (Izotóp Intézet, Budapest, Hungary) was added to the plates for 100 min. After scraping the cells into the medium, they were centrifuged for 5 min at 1000 rpm. We dissolved the sediment in 70 μL of lysis buffer, then glucose uptake was determined by scintillation counting by measuring 30 μL of the sample, using a Beckman LS 5000 TD counter, in counts per minute (CPM), for five minutes each, with average activity was used as the outcome. Following freezing overnight at −70 °C, protein concentration was measured with a Hitachi spectrophotometer. Results were normalized for protein content [44].

### 2.3. HPLC Analysis

The type of tyrosine aimed to be measured determined the sample preparation. Methods were based on earlier publications with minor modifications [39].

To measure the total intracellular non-protein-bound tyrosine concentration, prior to the freezing of samples overnight at −70 °C to achieve cell lysis, 200 μL distilled water was added to each. After melting up, samples were centrifuged for 15 min at 15,000 rpm. A total of 200 µL of the supernatant was mixed with 200 µL of 60% trichloroacetic acid. After 30 min incubation on ice, samples were centrifuged again at 15,000 rpm for 15 min. Then, the supernatant was filtered and was diluted 5-fold and then 160 µL distilled water was added to 40 µL of filtrate, followed by the injection of the mixture onto the HPLC column.

The total protein-bound cellular tyrosine content was measured by adding 200 μL of distilled water to the samples, followed by freezing overnight at −70 °C to achieve cell lysis. After melting and centrifugation for 10 min at 4000 rpm, 200 μL supernatant was mixed with 200 μL 60% trichloroacetic followed by incubation on ice for 30 min to precipitate proteins. After the second centrifugation for 10 min at 4000 rpm, the sediment was resuspended in 1% trichloroacetic acid and 4 µL of 400 mmol/L desferrioxamine. A total of 40 µL of 500 mmol/L butylated hydroxytoluene was added to the samples to avoid possible free radical formation during hydrolysis. Then, 200 µL of 6 N hydrochloric acid was added in order to hydrolyze the proteins at 120 °C overnight. The hydrolysate was then filtered through a 0.2 µm filter (Millipore Co., Billerica, MA, USA) and 20 µL of the filtrate was injected onto the HPLC column of a Shimadzu Class LC-10 ADVP HPLC system (Shimadzu USA Manufacturing Inc., Canby, OR, USA) using a Rheodyne manual injector.

The amounts of p-, o-, m-Tyr and Phe in the samples were determined by measuring their autofluorescence. Thus, no derivatization or staining was needed. Samples were measured on a Shimadzu Class 10 HPLC system equipped with an RF-10 AXL fluorescent detector (Shimadzu USA Manufacturing Inc., Canby, OR, USA). The mobile phase consisted of 1% sodium acetate and 1% acetic acid dissolved in water. The separation took place on a LiChroCHART 250-4 column (Merck KGaA, Darmstadt 64271, Germany) in an isocratic run. Wavelengths of 275 nm for excitation and 305 nm for emission were used to assess p-, o- and m-Tyr, while Phe was detected at 258 nm excitation and 288 nm emission wavelengths. Determination of the area under the curve (AUC) plus external standard calibration was used to calculate the precise concentrations of the amino acids.

### 2.4. Western Blot Analysis

The samples of lysates were vortexed and centrifuged (10 min, 13,000 rpm, at 4 °C). Protein content of the samples was determined by the Lowry method using bovine serum albumin as a standard. Samples were solubilized in 100 mmol/L Tris-HCl (pH 6.8), 4.0% sodium dodecyl sulphate (SDS), 20% glycerol, 200 mmol/L DTT, and 0.2% bromophenol blue containing buffer. Samples (80 to 120 μg protein) were electrophoretically resolved on 7.5% polyacrylamide gels and transferred to PVDF membranes (Amersham-Biotech, AP Hungary, Budapest, Hungary). Membranes were stained with Ponceau dye to ensure successful transfer. The non-specific antibody binding sites were blocked in 5% BSA in TBS-T solution at room temperature for one hour. Membranes were incubated in the primary antibodies anti-phospho-(Ser473)-Akt (Cell Signaling Technology, #7074, Beverly, CA, USA), to detect p-Akt in a final dilution of 1:2000, or anti-phospho-(Tyr612)-IRS-1 (1:2000 I2658 Sigma, Budapest Hungary) and they were used overnight at 4 °C. Membranes were washed three times for 5 min with TBS-T and incubated with HRP-conjugated anti-rabbit IgG secondary antibody (#7074, Cell Signaling) diluted in the blocking solution (1:4000) for one hour at room temperature. Membranes were washed three times for 5 min with TBS-T.

To re-probe Western blots with alternative primary antibodies, the stripping of membranes took place as follows: they were washed in 0.1% TBS-T for 10 min, the membranes were merged in stripping buffer containing 0.1% SDS, 1.5% glycine, and 1% Tween-20 at pH 2.2, twice for 10 min, then were washed in PBS twice for 5 min, and then in TBS-T 0.1% twice for 5 min. The membranes were blocked in 5% BSA in TBS-T solution at room temperature for one hour and then incubated with the following antibodies: total PKB/Akt in 1:1000 final dilution (#9272, Cell Signaling Technology, Beverly, MA USA) and total IRS-1 in 1:1000 final dilution (I7153 Sigma) overnight at 4 °C. The membranes were washed three times for 5 min with TBS-T and incubated with HRP-conjugated secondary antibody (#7074 Cell Signaling Technology, Beverly, MA USA) diluted in the blocking solution (1:2000) for one hour at room temperature. Membranes were washed three times for 5 min with TBS-T. Afterwards, membranes were incubated in enhanced chemiluminescence HRP substrate (ECL; Pierce Biotech, Bio-Rad, Budapest, Hungary) according to the manufacturer’s instructions. Computerized densitometry (integrated optical density) of the specific bands was analyzed using Scion Image for Windows software. Protein signals were corrected for total Akt or total IRS-1 protein levels and adjusted to controls.

### 2.5. Protein Expression

The cDNA segment encoding the human PTP1B catalytic domain (1–299 aa) was PCR amplified from a HEK293T cDNA pool and inserted into the pBH4 vector with BamHI and NotI restriction sites. This construct is only missing the C-terminal flexible region and the membrane binding segment of PTP1B. The protein was expressed with an N-terminal hexa-histidine tag. Recombinant active PTP1B was expressed in the Escherichia coli BL21 (DE3) bacterial strain. Briefly, cells were grown in an ampicillin-containing LB medium at 37 °C to OD = 0.6 and then cooled down to 18 °C and induced by the addition of 0.05 mM isopropyl-beta-D-thiogalactoside (IPTG). After an overnight expression at 18 °C, the pellet was harvested and washed with phosphate buffered saline (PBS). Following freezing at −80 °C, cells were lysed in an appropriate buffer (300 mM NaCl, 50 mM phosphate, 10 mM imidazole, 2 mM beta-mercapto-ethanol, 0.1% IGEPAL with pH = 8.0, and 0.5 mM benzamidine with 0.5 mM PMSF protease inhibitors added) with the help of sonification. The lysate was centrifuged at 20,000 rpm for 30 min and the supernatant (~30 mL) was mixed with 1 mL Ni-NTA resin slurry (50%). After 45 min incubation at 4 °C, the mixture was transferred to gravity columns and washed with 10-10 mL of imidazole (40 mM imidazole, 300 mM NaCl, 50 mM phosphate, pH = 8.0 with 2 mM beta-mercapto-ethanol) and high salt (1000 mM NaCl, 20 mM imidazole, 20 mM TRIS, pH = 8.0 with 2 mM beta-mercapto-ethanol) containing wash buffers, each. PTP1B was eluted by applying 5 mL elution buffer (400 mM imidazole, 200 mM NaCl, 10% glycerol, 20 mM TRIS, pH = 8.0, with 0.1% IGEPAL). The eluted protein was supplemented with tricarboxy-ethyl-phosphine (TCEP) reducing agent at 2 mM concentration. 

The Ni-NTA purified PTP1B protein was further purified by anion exchange chromatography. After an overnight dialysis against 1 L buffer with low salt (5 mM NaCl, 10% glycerol, 20 mM TRIS, pH = 8.0 with 1 mM dithiothreitol reducing agent), it was loaded on a resource Q anion exchange column and subjected to a gradient from 5 mM to 1 M NaCl. The protein practically eluted in a single peak, which was then pooled, supplemented with a reducing agent (TCEP), and frozen on liquid nitrogen. Final protein samples checked by SDS-PAGE were found to have a purity over 95%.

The N- and C-terminal SH2 domains (regions 321–433 and 614–724, respectively) of the PI3K regulatory subunit were subcloned into a bacterial expression vector by PCR and then similarly expressed and purified as described above.

Enzymatically active IR kinase (989–1382, fused to GST and produced in SF9 cells) was ordered from SinoBiological (catalog number 11081-H20B1). The stock solution was aliquoted and frozen separately to preserve activity. Prior to comparative measurements, the activity of recombinant IR kinase was tested on internal control peptides and found to be suitable for kinase assays.

### 2.6. Peptide Synthesis

Peptides conferring to one of the YxxM motif regions (626-GRKGSGDYMPMSPKV-639) of human IRS-1 were chemically synthesized. The Y abbreviation denotes normal, ortho, and meta tyrosine. The solid-phase peptide syntheses were performed using a Liberty Microwave Peptide Synthesiser (CEM Corporation, Matthews, NC, USA) applying the Fmoc/tBu strategy. The resin used was PL-Rink-Amide MBHA. The Fmoc group was removed by 4.5 equiv. piperazine/HOBtxH_2_O in DMF. The coupling steps were performed with 4 equiv. of Fmoc amino acids in DMF, 4.5 equiv. of HOBt/HBTU in DMF and 10 equiv. DIPEA in NMP, using a microwave power of 25 W for 2 × 300 s. All couplings were performed at standard double coupling conditions at a maximum temperature of 75 °C, except for the following amino acids: 2-Fmoc-amino-3-(3-hydroxyphenyl)-propanoic acid, 2-Fmoc-amino-3-(2-hydroxyphenyl)-propanoic acid, and 2-Fmoc-amino-3-(4-hydroxyphenyl)-propanoic acid (Fmoc-tyrosine), which were double coupled using a 25 min coupling followed by a 5 min period at 25 W. The phenolic hydroxyls were unprotected. Cleavage of the peptides from the solid support was carried out using a TFA containing cocktail with TFA 90%, water 5%, TIS 2.5%, and DTT 2.5%. Conditions: 10 mL cocktail, 3.5 h reaction time, RT. The resulted crude peptides were filtered, and the filtrates were lyophilized. The peptides were analyzed on an Agilent 1200/Waters SQD RP-HPLC/MS instrument, applying Phenomenex Proteo 4μm C18 90 Å column (4.6 × 250 mm) at 1 mL/min flow using a liner gradient of 5% to 85% B over 25 min. The solvent system used was A (0.1% TFA in H_2_O) and B (0.1% TFA in MeCN). The crude peptides were purified on a semipreparative RP-HPLC Shimadzu instrument applying a Phenomenex Luna 10 μm C18 100 Å 10 × 250 mm column.

For the synthesis of the phosphorylated derivatives, first, the C-terminal octapeptides were prepared in a manner similar to the previous ones. After the incorporation of the tyrosine moiety, the phosphorylation was made on-line, so the elongation of the peptide chains was stopped. The phosphorylation of the hydroxyl unprotected tyrosines were carried out on solid-phase using 10 equiv. di-tert-butylN,N-diethylphosphoramidite, 20 equiv. 1H-tetrazole, THF, followed by oxidation using 14% tert-butyl hydroperoxide/water. After the formation of the appropriately protected phosphotyrosine-containing peptides, the elongation of the peptide chains was continued and the remaining 7 amino acids were incorporated as described previously. Cleavage from the solid support and the processing of the crude peptide was performed using the above-mentioned methodology.

The two methionines in the sequence during the oxidation of the phosphite to phosphate were converted to methionine sulfoxide. Therefore, an additional step was necessary to transform the sulfoxides back to thioether. The reduction of the sulfoxides was performed in the solution phase using 20 equivalents of NH_4_I in TFA/H_2_O (1:1 *v*/*v*%) at 0 °C. It was particularly fast, with 100% conversion to the desired peptides after 30 min. The resulting phosphopeptides were purified as mentioned above. The HPLC characterization is in Table 1. 

Mass spectrometry: calculated Mw 1608.89, measured Mw 1608.6 (nonphosphorylated peptides); calculated Mw 1688.89, measured Mw 1689.0 (phosphorylated peptides).

### 2.7. Phosphorylation and Dephosphorylation Assays—Capillary Electrophoresis

The kinase assay mixture contained 200 μM peptide (GRKGSGDYMPMSPKV) and 875 nM IR kinase in kinase buffer (20 mM potassium phosphate (monobasic), 15 mM sodium phosphate (dibasic), 103 mM NaCl, 5 mM MgCl_2_, 5% glycerol, 0.05% Igepal; pH 7.5). The reaction was initialized by addition of 1 mM ATP. The phosphatase reaction mixture contained 80 μM peptide (GRKGSGD(phosphoY)MPMSPKV) in buffer (50 mM TRIS, 150 mM NaCl, 1 mM EDTA and 2 mM DTT; pH 7.4). The reaction was initialized by addition of 25 nM PTP1B enzyme. Equal aliquots of the kinase or the phosphatase reaction mixture were subjected to capillary electrophoresis analysis at the indicated time points after the start the reaction.

Background electrolyte (BGE) components phosphoric acid and triethylamine were purchased from Sigma (St. Louis, MO, USA) and from Merck GmbH (Darmstadt, Germany), respectively. Capillary electrophoresis was performed with an Agilent Capillary Electrophoresis 3DCE system (Agilent Technologies, Waldbronn, Germany) applying DB-WAX coated capillary having a 33.5 cm total and 25 cm effective length with 50 μm I.D. (Agilent Technologies, Santa Clara, CA, USA). On-line absorption at 200 nm was monitored by DAD UV-Vis detector. The capillary was thermostated at 25 °C. Before measurements, the capillary was rinsed subsequently with distilled water for 15 min and between measurements with BGE (100 mM trimethylamine-phosphate buffer; pH 2.5) for 3 min. Samples were injected by 5 × 10^3^ Pa pressure for 6 s. Runs were performed in the positive-polarity mode with 20 kV. 

### 2.8. Protein-Peptide Binding Assays

For in vitro fluorescence polarization (FP)-based affinity measurements, a peptide containing the phosphorylated tyrosine residue 632 of human IRS1 was labelled by lys-carboxyfluorescenine at its C-terminal end (GRKGSGD(pY)MPMSPKS(K-FITC). Synthesis of the labelled peptide was done by GenScript Inc. For competitive FP measurements, unlabeled peptides with either natural (para-phospho-Tyr) or unnatural (ortho-phospho-Tyr and meta-phospho-Tyr) phosphorylated amino acids (core sequence GRKGSGD(pY)MPMSPKV in all three cases) were used. Direct titration with the labelled peptide was done by generating a dilution series of the protein (N-terminal and C-terminal SH2 domains of PI3K regulatory subunit 1), with a constant (100 nM) peptide concentration in a standard buffer (100 mM NaCl, 20 mM TRIS pH = 8.0, 0.05% Brij-35 detergent). Competitive titrations were done by keeping the concentration of the protein-labelled peptide mixture constant (150 nM and 1500 nM C-terminal and N-terminal domain, respectively, with 100 nM labelled peptide) and varying the concentration of the unlabeled competitor peptide instead. Measurements were done in a Cytation C3 (BioTek Instruments) plate reader in black 384-well plates, using three parallels for all points. The resulting curves were fitted with OriginPro 7 (Origin Labs Inc., Wellesley, MA, USA). 

### 2.9. Immunofluorescence

In a 6-well plate, 104 cells were cultured on glass coverslips washed with alcohol and dried under UV light. The podocyte cells were grown in RPMI1640 medium supplemented with 10% heat-inactivated FBS at 33 °C, 5% CO_2_. When they reached 60% confluence, the medium was changed to RPMI1640 medium containing 2% FBS and 112.05 nM of the different tyrosine isoforms, then the plates were transferred to 37 °C, 5% CO_2_ to allow the cells to differentiate. The different tyrosine isoforms were added to the maintenance medium. The cells were serum deprived overnight, then incubated with 400 nmol/l insulin for 10 min. After removal of the medium, the coverslips were washed twice with PBS. The cells were fixed at room temperature in 2% paraformaldehyde and 4% sucrose for 8 min, then permeabilized using 0.3% Trion X-100 in 1xPBS for 20 min and blocked in 2.5% BSA for 45 min [45]. The primary antibody was diluted in 1xPBS, and the cells were incubated in it for 60 min in humid conditions. The antibodies we used were mouse anti-WT1 antibody (H-1) (1:100, Santa Cruz Biotechnology, Dallas, TX, USA), chicken anti-vimentin antibody (1:100, Abcam), rabbit anti-glucose transporter GLUT4 antibody (1:100, Abcam, Cambridge, UK), mouse Insulin Receptor Substrate-1 antibody (1:100, Invitrogen, Waltham, MA, USA), and rabbit anti-phospho-Insulin Receptor Substrate-1 (pTyr612) antibody (1:10, Sigma). Coverslips were washed in PBS three times for 5 min in PBS and incubated in the fluorophore-conjugated secondary antibodies anti-goat, anti-chicken, secondary antibody Alexa fluor 647, anti-rabbit Alexa fluor 350, and anti-mouse Alexa Fluor 488 (1:10, Invitrogen) for 60 min. Samples were then washed in PBS three times for 5 min and mounted in Vectashield (Vector Laboratories). Images were taken with a Nikon Eclipse Ti2 microscope.

### 2.10. Statistical Analysis

Statistical analysis was carried out using the SPSS Statistics 27 (IBM Company, Chicago, IL, USA) and GraphPad Prism vs 8 software (GraphPad Software, San Diego, CA, USA) packages. Experiments were carried out in replications up to n = 5–10 for certain experiments. The data obtained were checked for normality of distribution using the Kolmogorov–Smirnov test. Data with normal distribution were analyzed using parametric tests, while non-parametric tests were used for non-normally distributed data. For multiple comparisons, analysis of variance (ANOVA) with post-hoc analysis was performed for normally distributed data. For non-normally distributed data, the Kruskal–Wallis test, and, upon significance, pairwise comparison with the Mann–Whitney U tests were carried out. Pairwise comparisons of normally distributed data were carried out using independent samples *t*-tests. If the control was set to 100% and different experimental setting were compared to that (e.g., Figures 3–5), a one-sample *t*-test was used.

## 3. Results

### 3.1. Ortho- and Meta-Tyrosine Inhibit Insulin-Induced Glucose Uptake

We first tested the glucose uptake of differentiated 3T3-L1 cells in the absence and presence of o-Tyr and m-Tyr in culture media containing normal or high concentrations of glucose. In cells grown on p-Tyr containing cell culture media, increasing concentrations of insulin led to a marked increase in glucose uptake in 5 mmol/L glucose, but not in 25 mmol/L glucose. Similarly to the high glucose environment, in cells grown in media supplemented with o- and m-Tyr increasing concentrations of insulin failed to induce the glucose uptake under normal (5 mmol/L) glucose concentrations (Figure 3).

The inhibitory effect of o- and m-Tyr was also tested for zero time-dependence (Figure 4). While in cells grown on p-Tyr insulin led to an approximately two-fold increase in glucose uptake, cells grown in media supplemented with o- and m-Tyr insulin showed no significant effect, and even non-stimulated (basal) glucose uptake was lower than that of the p-Tyr control. Importantly, cells grown on o- and m-Tyr displayed similar deficiency in response to insulin under low and high glucose conditions alike, after one day or up to as long as twelve days (Figure 4).

The inhibitory effect of o-, and m-Tyr on the glucose uptake was ameliorated by increasing concentrations of p-Tyr (Figure 5).

#### 3.1.1. Both o- and m-Tyr Can Be Taken Up by Fat Cells within Several Minutes and Are Incorporated into Cellular Proteins 

Abnormal amino acids may alter insulin signaling by incorporation into proteins, which requires the cellular uptake of these amino acids. Therefore, we tested if the abnormal amino acids could be taken up by the cells. For that reason, non-protein-bound intracellular p-Tyr content, as well as the o-Tyr/p-Tyr and m-Tyr/p-Tyr ratios were measured. We found a continuous uptake of amino acids which was independent of glucose concentration (5 or 25 mmol/L) and the presence of insulin in the medium (Figure 6).

In a long-term experiment, we tested whether o- and m-Tyr are incorporated into cellular proteins. In cells grown on o-Tyr or m-Tyr, the p-Tyr/Phe ratio showed either a decrease or remained unchanged (Figure 7A,D,G,J), while protein-bound o-Tyr/p-Tyr and m-Tyr/p-Tyr increased (Figure 7B,C,E,F,H,I,K,L).

#### 3.1.2. Phosphorylation of IRS-1 in Cells Grown on o- or m-Tyr

In order to elucidate the mechanism underlying the inhibitory effect of o- and m-Tyr on insulin-induced glucose uptake, the phosphorylation levels of the insulin-receptor substrate-1 (IRS-1) and Akt (protein kinase B) steps of insulin signaling responsible for glucose uptake were studied. We found that in p-Tyr containing media, insulin treatment led to an approximately two-fold increase in the activating phosphorylation of IRS-1 and Akt. Interestingly, in cells grown on o- and m-Tyr containing media, the basal levels of IRS-1 and Akt phosphorylation were either unchanged or higher, which, however, could not be further raised by insulin treatment (Figure 8).

### 3.2. Biochemical Characterization of IRS Peptides Containing Different Forms of Tyrosine

Insulin receptor (IR) kinase phosphorylates multiple tyrosine residues in IRS-1 and its phosphorylation plays a central role in mediating signals towards downstream targets [46]. IRS-1 binds to activated IR kinase by its phospho-tyrosine binding (PTB) domain, and its plekstrin homology (PH) domain is instrumental to its cell membrane binding. In addition, this adapter protein contains a long, disordered C-terminal tail containing six YXXM motifs phosphorylated by the IR kinase, and thus provides a versatile contact for tyrosine phosphorylation dependent recognition of multiple regulator and effector proteins involved in the insulin pathway (Figure 9A). In addition to being the major substrate sites for the IR kinase, phosphorylated YXXM motifs are dephosphorylated by protein tyrosine phosphatase 1B (PTP1B) [47]. Furthermore, phosphoinositide 3-kinase (PI3K) is recruited to the IR kinase signaling complex at the cell membrane via the Src homology 2 (SH2) domains of its p85 regulatory subunit [48].

In order to address the putative roles of o- and m-tyrosine incorporated into IRS-1, we studied the phosphorylation and the dephosphorylation of IRS-1 YxxM motif containing peptides by IR kinase and PTB1B, respectively. Peptides, corresponding to a fifteen amino acid long YXXM motif containing IRS1 (region 626–639) fragment, were chemically synthesized with unphosphorylated or phosphorylated p-, o- or m-tyrosines. The phosphorylation state of these peptides was examined by capillary electrophoresis in an in vitro kinetic experiment. In this assay, the electrophoretic mobility of peptides changed according to the tyrosine phosphorylation state. We found that neither o- nor m-tyrosine containing peptides are substrates for the IR kinase, while the enzyme efficiently phosphorylated the p-Tyr containing “natural” peptide (Figure 10A). Similarly, apart from the p-Tyr peptide, none of the modified Tyr containing peptides could be dephosphorylated by PTP1B (Figure 10B).

In order to study the role of o- and m-Tyr amino acid incorporation regarding other IRS-1 partner proteins relaying insulin signals, the binding capacity of para-, ortho-, and meta-tyrosine containing IRS-1 peptides to PI3K SH2 domains were also investigated (the regulatory subunit contains an N- and C-terminal SH2 domain). Using an in vitro protein-peptide binding assay, we found that o- and m-Tyr containing peptides had greatly reduced binding affinity to these SH2 domains (Figure 11). These in vitro protein–peptide binding and enzyme activity results can be structurally explained by observing the crystal structures of known protein–peptide complexes containing natural, p-Tyr, or phospho-p-Tyr possessing peptides: the topology of the binding surface on the interacting protein is only compatible with p-Tyr or phospho-p-Tyr for IR kinase or PTB1B and SH2 domains of PI3K, respectively (Figure 9B).

The panels below show that para-tyrosine containing peptides are efficiently phosphorylated and dephosphorylated by IR kinase and PTP1B, respectively, while meta- and ortho-tyrosine peptides are very poor substrates of these enzymes.

Results of the fluorescence polarization (FP)-based binding assays. To the left, the panels show the direct titration of the fluorescently labelled para-tyrosine containing phosphorylated IRS-1 peptide with the isolated N-terminal and C-terminal SH2 domains of the PI3K regulatory subunit (top and bottom, respectively). The next three panels on the right show the competitive binding curves with increasing amounts of phospho-para- (GRKV2), phospho-meta- (GRKV4), and phospho-ortho-Tyr (GRKV6) containing peptides. Notice that both modified phospho-tyrosine containing peptides (GRKV4 and 6) have greatly reduced binding affinity to the SH2 domains compared to the phospho-para-Tyr containing peptide (GRKV2). Note the change from nanomolar to micromolar binding affinity.

### 3.3. Microscopical Analysis

Visualization of insulin signaling is shown in Figure 12 and Figure 13. IRS-1 phosphorylation leads to membrane translocation of this signaling protein, which is clearly demonstrated by Figure 12 in p-Tyr containing medium, but this translocation is absent in the presence of m- and o-Ty. Moreover, the intensity of p-IRS-1 is also lower in the case of m- and o-Tyr. The same membrane translocation on insulin treatment is also characteristic of GLUT-4, which is responsible for insulin dependent glucose transport. Figure 13 shows this characteristic localization of GLUT-4 positivity in the presence of p-Tyr, which is absent in samples treated with m- and o-Tyr.

## 4. Discussion

In this study, we report for the first time, that the abnormal amino acids, o- and m-Tyr inhibit glucose uptake of fat cells. Furthermore, we provide evidence that insulin signaling may be altered in cells grown on media containing o- or m-Tyr. We also observed that o- and m-Tyr can be taken up by the cells and incorporates into cellular proteins.

It is suggested that there is a causal connection between oxidative stress and insulin resistance, e.g., oxidized LDL, as well as isoprostanes correlated with insulin resistance as measured by the homeostasis model assessment index (HOMAIR) [53]. On the other hand, oxidative stress has been shown to activate serine protein kinases that would interfere with insulin signaling [23]. Oxidative stress is believed to lead to the activation of inflammatory processes that could further contribute to the development of insulin resistance through inflammatory cells [4] or via activation of stress-kinases (e.g., JNK), which could lead to the serine/threonine (Ser/Thr) phosphorylation of IRS-1 and IRS-2 and in this way impair insulin signaling [54].

The abnormal amino acids o- and m-Tyr are results of the attack of hydroxyl radicals on Phe molecules or the Phe residues of proteins [55]. o- and m-Tyr are regarded as specific, stable markers of hydroxyl radicals [56], and have been detected in increasing amounts in cataract lenses [38], in urine of preterm infants [57], in Fabry’s disease [58], and in cardiopulmonary bypass [59], among others. The abnormal tyrosine isomers offer the advantage of fluorescent detection without derivatization upon their autofluorescence [60].

Our data suggest that while in cells grown on p-Tyr containing culture medium, insulin is able to induce glucose uptake to approximately two-fold; in cells grown on media enriched with glucose, o-, or m-Tyr, glucose uptake is similarly blunted. The effects can already be observed after a single day, and last at for least up to 12 days. These observations prompt for a direct role of o- and m-Tyr, and not the role of oxidative stress itself, as in this setting o- and m-Tyr were applied in the absence of an obvious oxidative stress.

Our present data suggest that supplementation of four different cell lines with o- or m-Tyr leads to insulin resistance in the cells, similarly to a high glucose environment. In the o- and m-Tyr grown cells, insulin failed to increase IRS-1 phosphorylation.

While insulin was able to stimulate Akt phosphorylation in p-Tyr grown cells in normal glucose media, in the presence of o- and m-Tyr, insulin was unable to exert any effect, similar to the high glucose environment.

From the insulin receptor substrate proteins, only IRS-1, but not IRS-2, was studied, and both would have an impact on Akt phosphorylation [61,62]. Furthermore, only one phosphorylation site of IRS-1 was investigated, although numerous Tyr phosphorylation sites exist [63]. Furthermore, incorporation of o- or m-Tyr into cellular proteins such as IRS-1 might alter immunologic recognition by antibodies, especially directed against phosphorylated tyrosine residues.

Moreover, other protein kinases, e.g., stress kinases (e.g., JNK) may influence insulin signaling at different levels [54,62], and hypothetically, these kinases could also be altered by the addition of o- or m-Tyr to the culture media.

Additionally, constant stimulation of insulin signaling has been shown to alter kinetics and extent of phosphorylation of IRS-1 and Akt and cause seeming disparities between them [64]. Nevertheless, the results of Akt phosphorylation and glucose uptake match each other well, indicating the development of insulin resistance at both levels upon o- or m-Tyr supplementation.

Biochemical characterization of a YXXM motif containing IRS1 peptide demonstrated that the position of the phosphorylatable hydroxyl group greatly affects its binding capacity to several IRS1 interactors (IR, PTP1B and PI3K-SH2). These interactors have binding grooves with distinct binding surface topographies varying from shallow (IR) to deep (PTP1B), but o- and m-Tyr were deleterious for binding in each case. This indicates that incorporation of these phenylalanine oxidization by-products into signaling proteins will greatly perturb network output and behavior because of the fundamentally altered protein–protein binding capacity of their components.

Altogether, we present a rather indirect connection between the incorporation of o-and m-Tyr into proteins and their effect on insulin signaling. Unfortunately, to date, there are no commercially available antibodies raised against o- and m-Tyr or o- and m-Tyr containing proteins. Moreover, the lack of isotope-labelled o- or m-Tyr precluded providing a more direct link. For the same reason, the incorporation of o- or m-Tyr into specific proteins, such as the insulin receptor IRS-1 or other signaling molecules, cannot be demonstrated. Furthermore, 3T3-L1 cells are not real fat cells, but fat cell-like cells; however, they provide a widely accepted model to study insulin signaling in fatty tissue.

In spite of these shortcomings, our study does suggest that o- and m-Tyr would be able to incorporate into cellular proteins in fat cells. Moreover, our results are consistent with a model in which these incorporated amino acids interfere with insulin signaling and inhibit insulin-stimulated glucose uptake, i.e., they induce metabolic insulin resistance in the cells, just like under hyperglycemic circumstances (Figure 14).

These results are in line with our previous observations regarding vascular insulin resistance [65], vascular liraglutide resistance [66], and EPO resistance [43], and suggest a potentially more universal role of o- and m-Tyr in the development of hormone resistances in conditions with high oxidative stress.

## Figures and Tables

**Figure 1 biomedicines-10-00975-f001:**
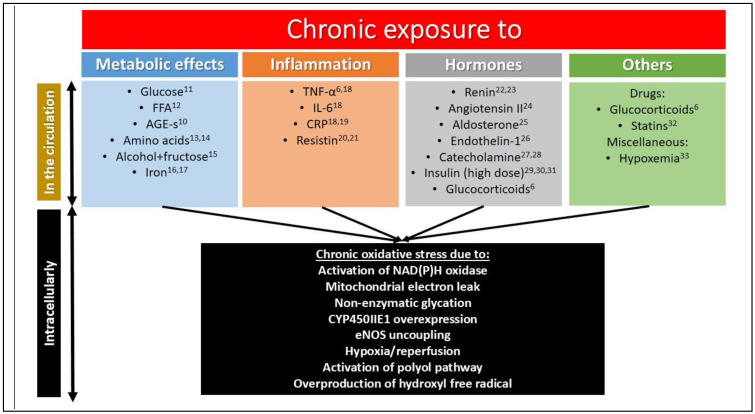
Chronic exposure leading to oxidative stress and insulin resistance [6,10,11,12,13,14,15,16,17,18,19,20,21,22,23,24,25,26,27,28,29,30,31,32,33]. Those factors are listed which cause oxidative stress and insulin resistance at the same time in a chronic setting. The arrows indicate a causal relationship between alterations in the circulation leading to intracellular abnormalities.

**Figure 2 biomedicines-10-00975-f002:**
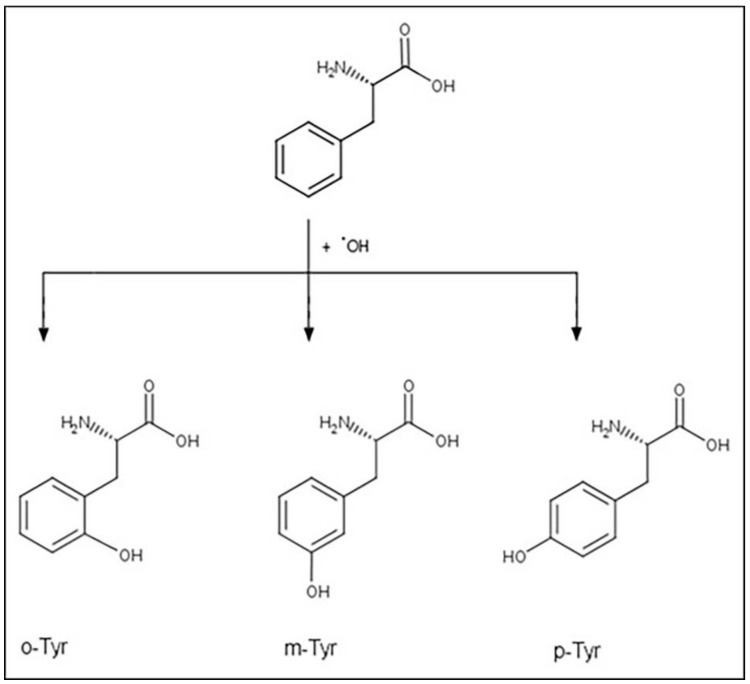
Hydroxyl free radical induced production of tyrosine isomers.

**Figure 3 biomedicines-10-00975-f003:**
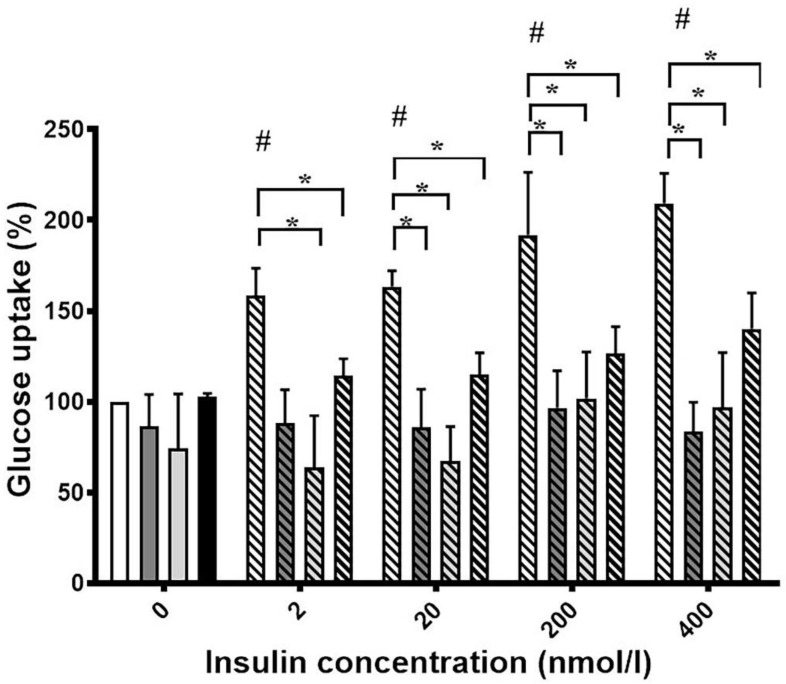
Insulin-dependent uptake of deoxy-D-glucose 2-[1 2-3H(N)] into differentiated 3T3-L1 adipocytes was assessed in media containing (i) para-tyrosine with 5 mmol/L glucose content (white column and first striated column in each block), (ii) meta-tyrosine with 5 mmol/L glucose (dark grey column and second striated column in each block), (iii) ortho-tyrosine with 5 mmol/L glucose (light grey column and third striated column in each block), and (iv) para-tyrosine with 25 mmol/L glucose content (black column and fourth striated column in each block). Cells were treated with 2, 20, 200, and 400 nmol/L insulin as shown (#, *p* < 0.05 vs. control para-tyrosine using one-sample *t*-test, *, *p* < 0.05 vs. 5 mmol/L glucose para-tyrosine using independent samples *t*-tests). Glucose uptake of untreated cells grown on para-tyrosine and 5 mmol/L glucose containing medium was set to 100%. Results are shown as a mean ± SEM for n = 5–10 individual measurements.

**Figure 4 biomedicines-10-00975-f004:**
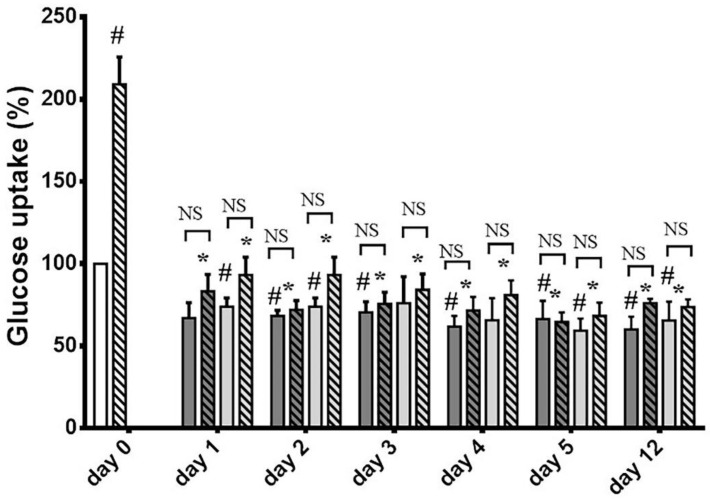
Insulin-dependent uptake of deoxy-D-glucose 2-[1 2-3H(N)] into 3T3-L1 adipocytes after cells were grown in media containing para-tyrosine meta-tyrosine or ortho-tyrosine for 1, 2, 3, 4, 5, or 12 days, with or without insulin treatment (200 nmol/L). The glucose uptake of untreated adipocytes grown on 5 mmol/L glucose medium containing para-tyrosine was set to 100%. Results are shown as a mean ± SEM for n = 5–8 individual measurements. #, *p* < 0.05 vs. para-tyrosine control (one-sample *t*-test); *, *p*< 0.05 vs. para-tyrosine + insulin (independent samples *t*-test), NS: non-significant. Bars indicate para-tyrosine (white column), meta-tyrosine (dark grey columns), ortho-tyrosine (light grey columns), control (simple columns), and insulin-treated (striated columns).

**Figure 5 biomedicines-10-00975-f005:**
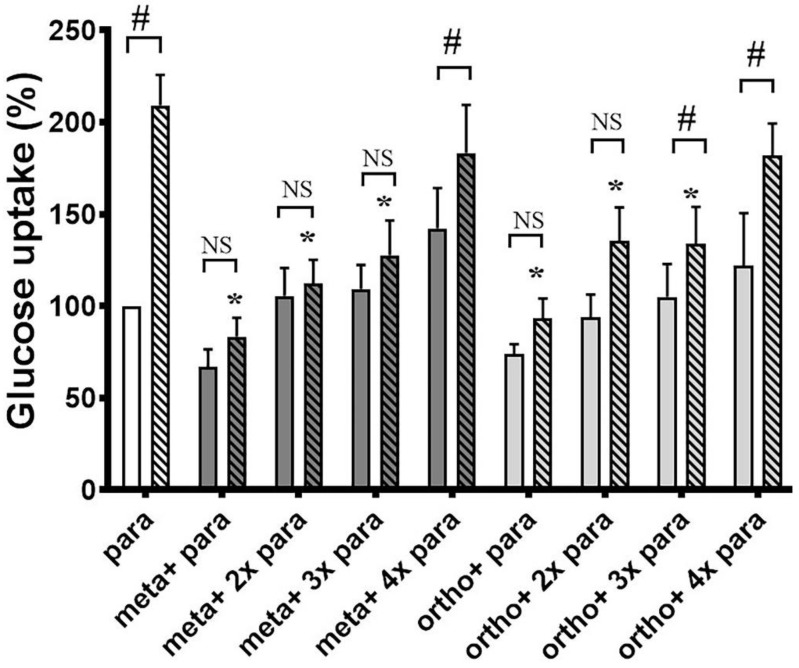
para-Tyr reverses the inhibitory effects of o- and m-Tyr. Examination of deoxy-D-glucose 2-[1 2-3H(N)] uptake of 3T3-L1 adipocytes depending on the ortho- and meta-tyrosine content of the medium in the absence of insulin (non-striated bars) or in the presence of 400 nmol/L insulin (corresponding striated bars). The basal glucose uptake of the cells, grown on the original 0.39 mmol/L para-tyrosine containing medium was considered to be 100%. Results are shown as mean ± SEM after n = 10 individual measurements. *, *p* < 0.05 vs. para-tyrosine + insulin (independent samples *t*-test), #, *p* < 0.05 vs. identical control (one sample *t*-test or independent samples *t*-test accordingly), NS: non-significant.

**Figure 6 biomedicines-10-00975-f006:**
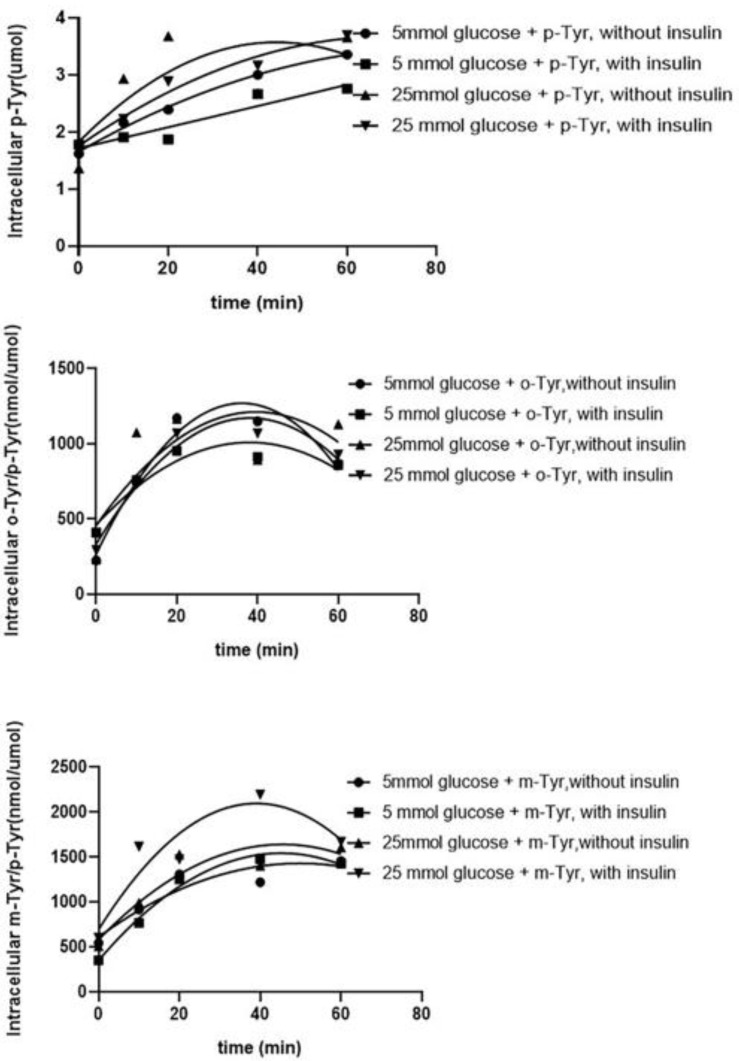
HPLC measurement of non-protein-bound, cytosolic, intracellular para- (upper panel), meta- (middle panel), and ortho-tyrosine (lower panel) content of 3T3-L1 adipocytes after time-dependent incubation with different tyrosines, without insulin, and after grown either in 5 mmol/L glucose containing medium or in 25 mmol/L glucose containing medium, or with insulin treatment (400 nmol/L), either in 5 mmol/L glucose containing medium or in 25 mmol/L glucose containing medium. Note that p-Tyr is shown as an absolute concentration, while o- and m-Tyr are depicted as their ratios to p-Tyr. There was no significant difference between the measurements (ANOVA).

**Figure 7 biomedicines-10-00975-f007:**
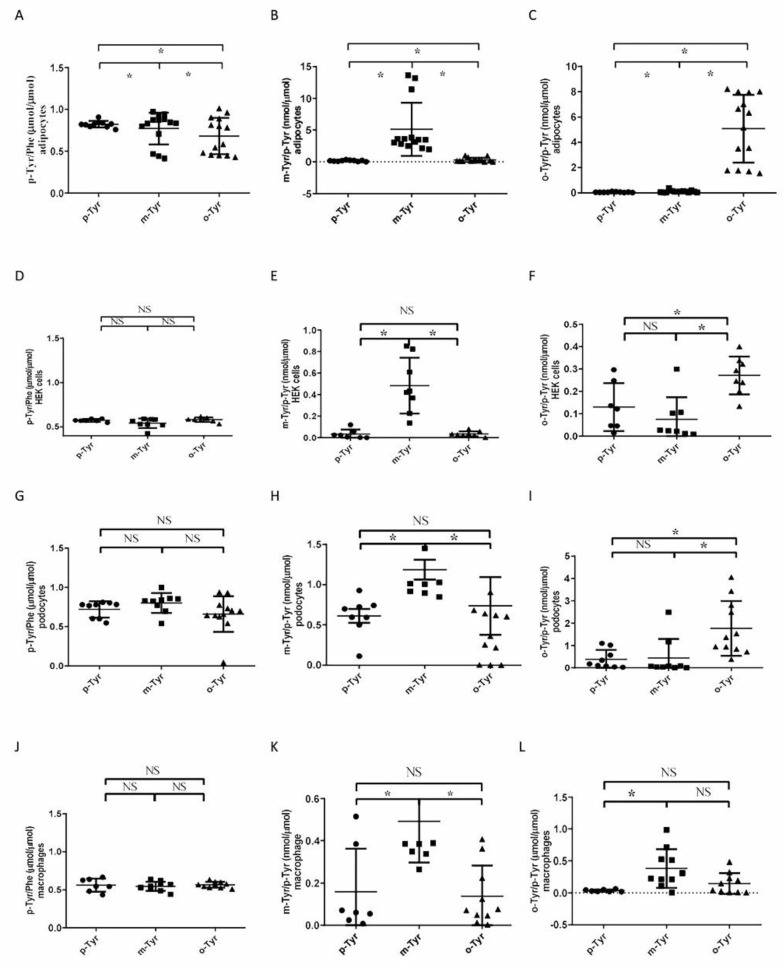
p-, m-, and o-Tyr content of the total proteins of cells grown in different Tyr media. HPLC measurement of protein-bound different tyrosine isomeres in cell lysates, grown in media containing para-, ortho-, or meta-tyrosine.* *p* < 0.05, NS: non-significant (using Kruskal–Wallis test for multiple comparisons and subsequently Mann–Whitney U test for pairwise comparison, as data were non-normally distributed). Results are mean ± SEM for n = 5–10 individual measurements. Note that the amount of p-Tyr is shown as p-Tyr/Phe ratio and is expressed in µmol/µmol units (panel **A**,**D**,**G**,**J**), while o- and m-Tyr are depicted as their ratio to p-Tyr (i.e., o-Tyr/p-Tyr and m-Tyr/p-Tyr, respectively) and the units are nmol/µmol (m-Tyr, panel **B**,**E**,**H**,**K**; o-Tyr, panel **C**,**F**,**I**,**L**).

**Figure 8 biomedicines-10-00975-f008:**
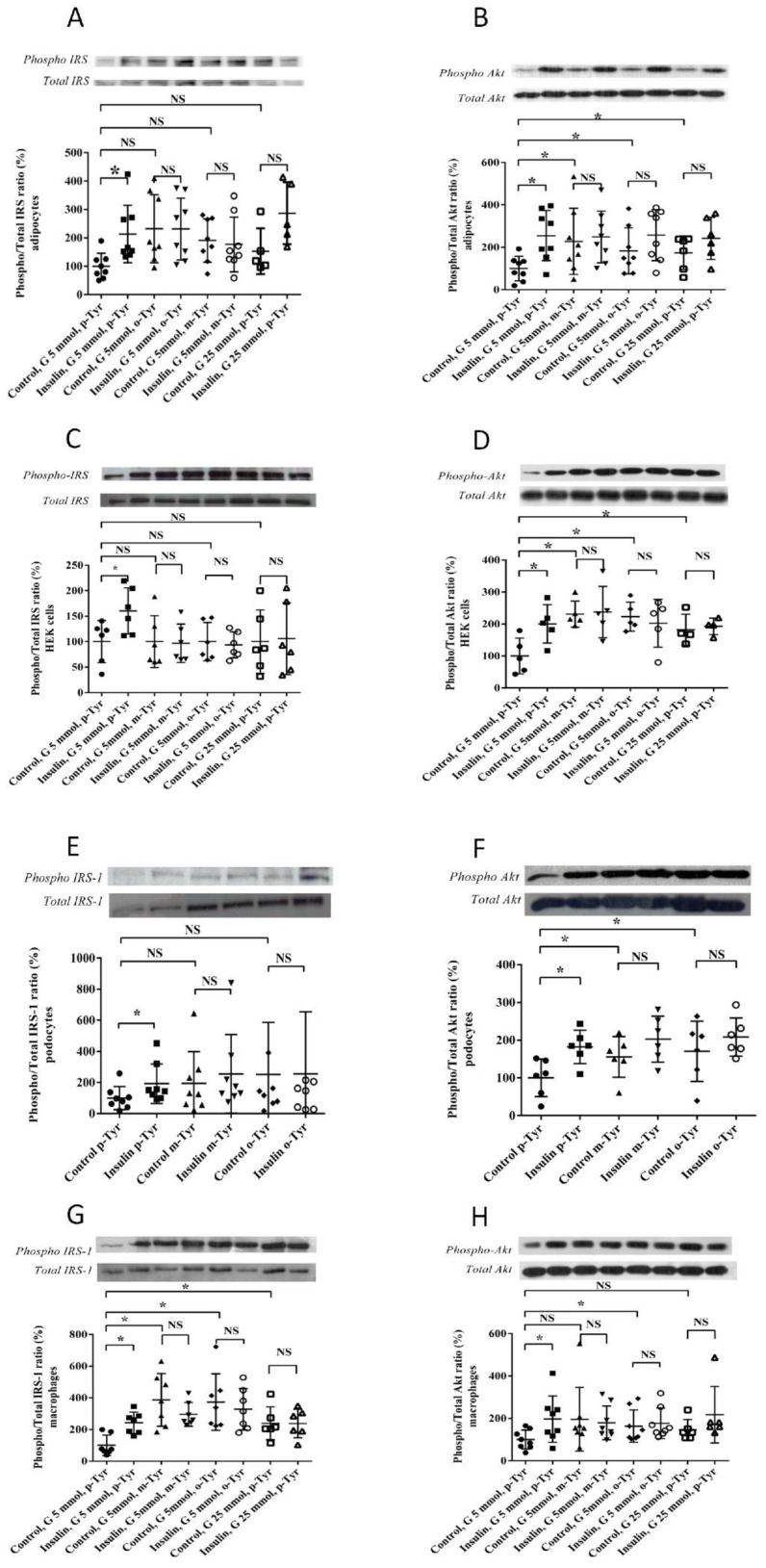
Western blot analysis of activating phosphorylation of IRS-1 (Tyr612, panel **A**,**C**,**E**,**G**) and Akt (Ser473, panel **B**,**D**,**F**,**H**) in the four cell lines (adipocytes, panel **A**,**B**; HEK cells panel **C**,**D**; podocytes, **E**,**F**; macrophages, panel **G**,**H**). Insulin-induced phosphorylation of IRS-1 (insulin receptor substrate-1) at the tyrosine of the first YXXM motif (Tyr612) in cells grown in media containing para-, meta-, or ortho-tyrosine with and without insulin treatment (400 nmol/L). Results are mean ± SEM for n= 4–8 individual measurements. * *p* < 0.05 (for non-normally distributed data, a Kruskal–Wallis test and, upon significance, pairwise comparisons with Mann–Whitney U test were carried out. Pairwise comparisons of normally distributed data were carried out using independent samples *t*-tests).

**Figure 9 biomedicines-10-00975-f009:**
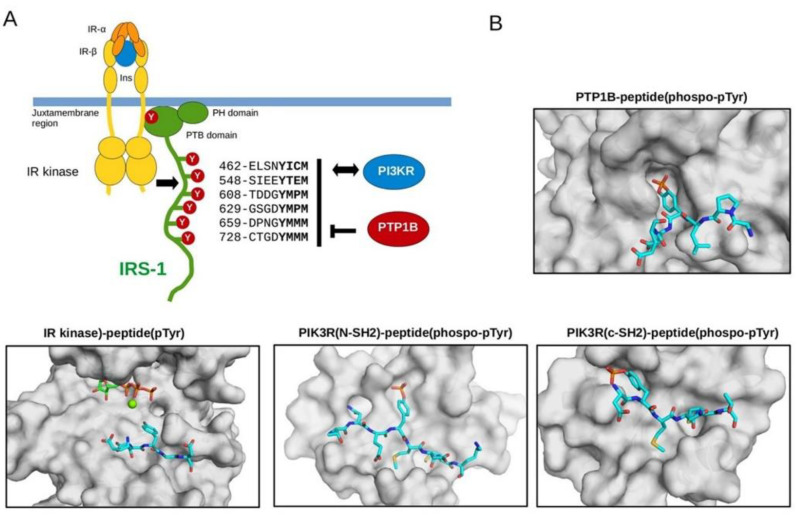
The role of IRS YXXM motifs in insulin mediated signaling. (**A**) Schematic of IR kinase mediated signaling: IR kinase, IRS-1 (PTB, PH domains and C-tail with YXXM motif positions and sequences indicated), PTP1B and PI3K regulatory subunits with SH2 domains. IRS-1 contains six YxxM motifs that play a central role in downstream signaling from the insulin receptor. The insulin receptor (IR)—comprised of two subunits—is dimerized upon binding to insulin (Ins). This activates the IR kinase which then creates a recruitment site in its juxtamembrane region for the PTB domain of IRS-1 by tyrosine phosphorylation. The PH domain helps recruiting IRS-1 to the cell membrane. (**B**) Crystallographic models of p-Tyr containing peptides binding to the deep substrate binding pocket of PTP1B, to the shallow substrate binding pocket of the IR kinase, and to the N-terminal SH2 domain of PI3K regulatory subunit (from left to right). IRS-1 partners are shown in surface representation, while substrate or ligand peptides from various proteins are shown with sticks. Structural figures were made by using the following protein–peptide PDB structures: 4zrt, PTP1B─Nephrin substrate phospho-peptide [49]; 3bu5, IR kinase IRS2─KRLB region peptide [50]; 2iuh, PI3KR-(N)SH2─c-Kit phospho-peptide [51]; 5aul, PI3KR-(C)SH2─CDC28 phospho-peptide [52].

**Figure 10 biomedicines-10-00975-f010:**
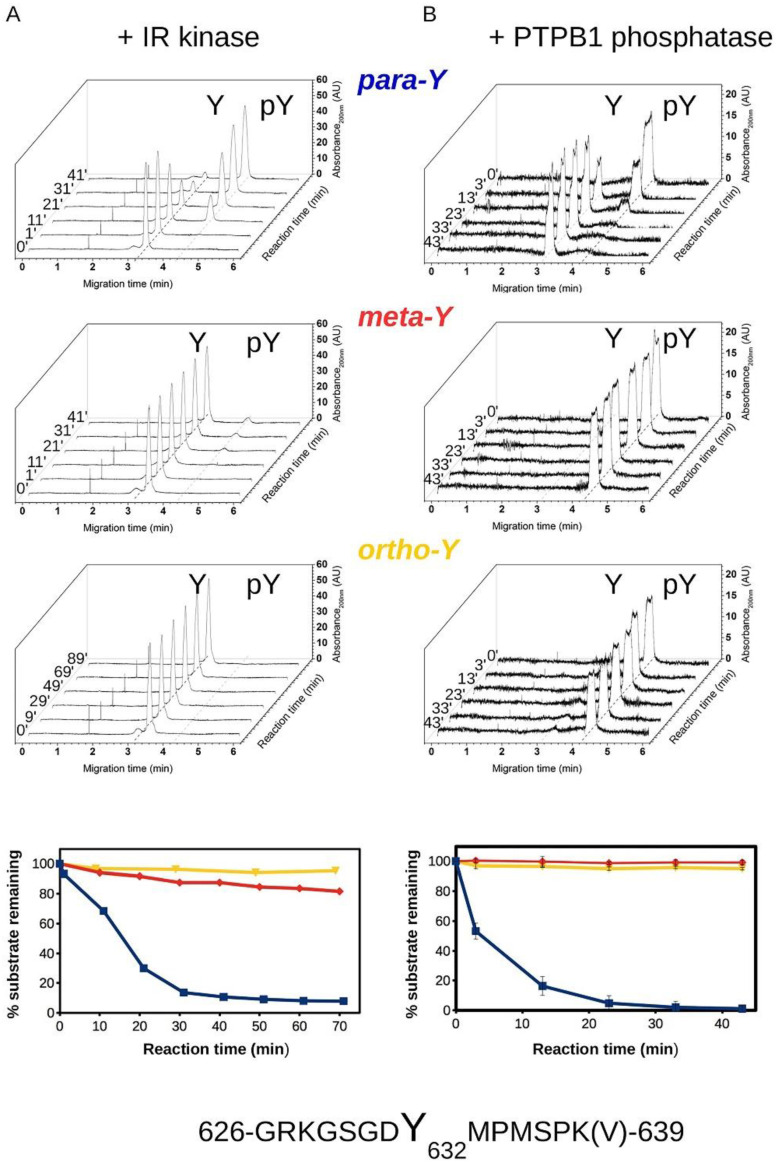
Phosphorylation and dephosphorylation of p-, m-, and o-Tyr containing IRS-1 peptides. (**A**) Results of in vitro kinase assays using recombinant IR kinase. Phosphorylation of IRS1 YXXM motif containing peptides with para-, meta-, and ortho-tyrosines was analyzed by capillary electrophoresis. After starting the reactions, sample aliquots were injected to the capillary at the indicated time points. Characteristic migration times for the unphosphorylated peptide are indicated by a dashed line (~3.2 min). Notice the appearance of a slower migrating peak (at ~4.2 min) corresponding to the phosphorylated peptide in the case of the peptide with para-tyrosine. (**B**) Results of PTP1B dephosphorylation assays. Experiments were performed similarly to kinase reactions, but the substrate peptides were previously phosphorylated. Characteristic migration times for the phosphorylated peptide are indicated by a dashed line (~4.2 min). Notice the appearance of a faster migrating peak (at ~3.2 min) corresponding to the dephosphorylated peptide in the case of the peptide with para-tyrosine.

**Figure 11 biomedicines-10-00975-f011:**
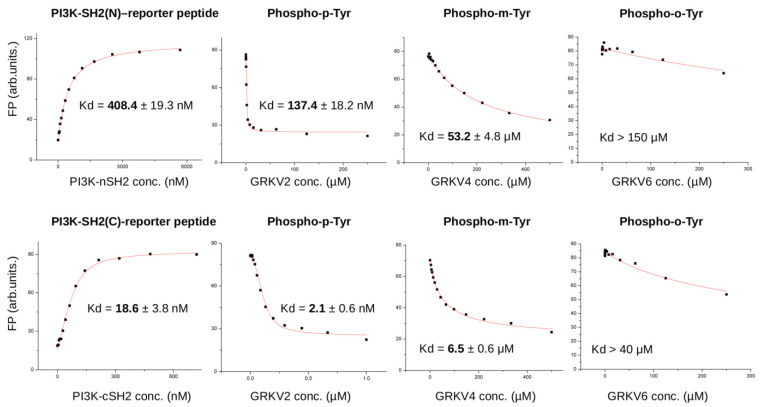
Binding of m- and o-Tyr IRS1 peptides to the SH2 domains of PI3K regulatory subunit. The error in the Kd values show the uncertainty of the numerical fit to the direct (**top**) and competitive (**bottom**) binding curves plotted as the mean of three technical replicates. For the GRKV6 peptide, the binding was so weak that its binding affinity could only be estimated. Please also take note that the binding affinity of the para-Tyr peptide is nanomolar (nM), while for the meta- and ortho-Tyr peptides this value is micromolar (μM), making statistical comparison unnecessary.

**Figure 12 biomedicines-10-00975-f012:**
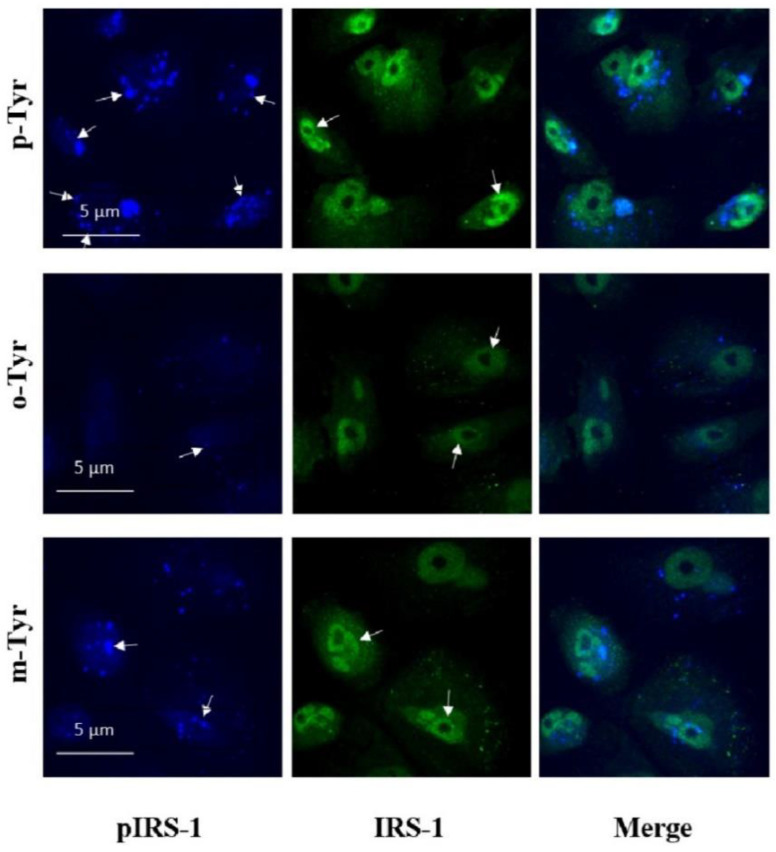
Immunofluorescence staining of insulin-treated podocytes for p-IRS-1 (blue) and total IRS-1 (green). p-IRS-1 (indicated by the white arrows) is located in the membrane when cells cultured in the presence of p-Tyr, which is not characteristic in cell treated with o- and m-Tyr. Moreover, p-IRS-1 is more intense in cells cultured in medium containing p-Tyr and the signal almost disappears in cells treated with o-Tyr. Total IRS-1 localization is mainly perinuclear.

**Figure 13 biomedicines-10-00975-f013:**
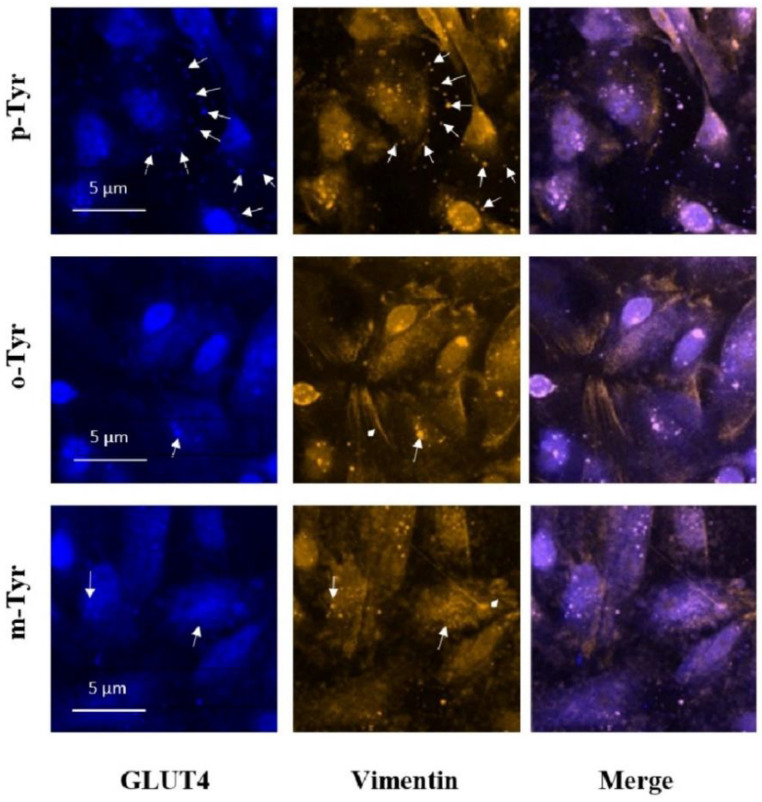
Immunofluorescence staining of insulin-treated podocytes for GLUT4 (blue) and vimentin (yellow). In p-Tyr-treated cells, GLUT4 aggregates are transported to the membrane. This localization is slightly visible in cells treated with m-Tyr, but it is not characteristic in o-Tyr treated cells, which shows rather a perinuclear localization. Typical localization of GLUT4 and vimentin is highlighted by the white arrows. Vimentin shows a colocalization with GLUT4. The arrowhead indicates the thicker vimentin filaments appearing in cells treated with m- and o-Tyr.

**Figure 14 biomedicines-10-00975-f014:**
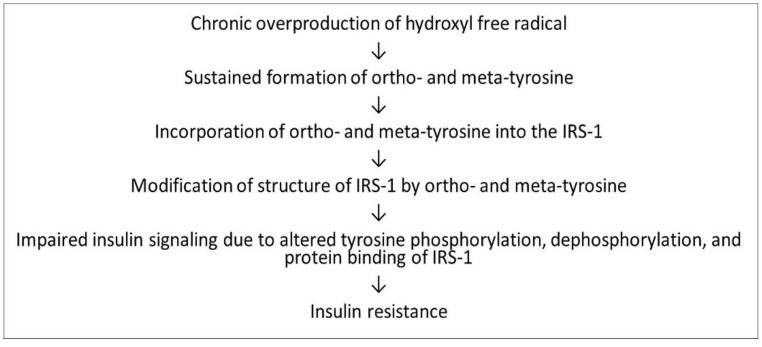
Suggested role of o- and m-Tyr in the development of insulin resistance.

**Table 1 biomedicines-10-00975-t001:** HPLC characterization of the peptides.

Peptide	t_R_ (min)
GRKGSGDF(4-hydroxy)MPMSPKV	t_R_ = 9.96
GRKGSGDF(3-hydroxy)MPMSPKV	t_R_ = 9.19
GRKGSGDF(2-hydroxy)MPMSPKV	t_R_ = 12.08
GRKGSGDF(4-PO_4_H_2_)MPMSPKV	t_R_ = 9.06
GRKGSGDF(3-PO_4_H_2_)MPMSPKV	t_R_ = 8.24
GRKGSGDF(2-PO_4_H_2_)MPMSPKV	t_R_ = 8.59

Gradient 20–35 % (B) in 15 min, flow 1.0 mL/min.

## Data Availability

Data presented in this study are available on request from the corresponding author.

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
