# Peer review of "Incorporation of Oxidized Phenylalanine Derivatives into Insulin Signaling Relevant Proteins May Link Oxidative Stress to Signaling Conditions Underlying Chronic Insulin Resistance"

_biomedicines, 2022, doi:10.3390/biomedicines10050975_

Round 1

Reviewer 1 Report

The paper “Incorporation of oxidized phenylalanine derivatives into insulin signaling relevant proteins may link oxidative stress to signaling conditions underlying chronic insulin resistance" by Cseh et al. deals with the role of 18 o-Tyr and m-Tyr in the development of insulin resistance.

The article is well written and only minor spell check is necessary. The paper has a good design. The article is logically divided into sections and subsections. The work has a good degree of novelty and of good interest to the readers.

Comments:

  • Introduction, 1.1. This section is too short. I suggest adding few more sentences. In particular, I suggest to further enhance the role of ROS generations induced by mitochondria, which is also the major cause of liver damage. “The increased lipid burden is also responsible for increased mitochondria activity (ββ-oxidation of free fatty acid, adenosine triphosphate (ATP) production and reactive oxygen species (ROS) generation) and mass. Over time, mitochondria may become exhausted, leading to uncoupling, with an increase in oxidative stress due to increased ROS formation and impaired hepatic insulin resistance” (doi: https://doi.org/10.31083/j.rcm2203082).

Reviewer 2 Report

The authors conduct a cell model and aimed to investigate that m- and o-Tyr –on the analogy of their potential role in EPO-resistance – are able to induce chronic insulin resistance in fat and HEK cells, in podocytes and macrophages.

Comments:

In this study, the authors are provided the P values and how they are used to determine true differences of outcome (glucose uptake) in tyrosine isomers groups (para-tyrosine ortho-tyrosine meta-tyrosine into 3T3-L1 adipocytes in Figures 1 to 3).

How to calculate the p values (statistical methods)?

Similarly, in Figure 6, P values should be provided; in Figures 7 and 8, how to calculate the P values?

In Figures 11, maybe the statistical relationships of binding of m- and o-Tyr IRS1 peptides to the SH2 domains of PI3K regulatory subunit should be provided.

Statistical analysis section should be provided in this manuscript.

Round 2

Reviewer 2 Report

No further comment